# A Random Forest Machine Learning Framework to Reduce Running Injuries in Young Triathletes

**DOI:** 10.3390/s20216388

**Published:** 2020-11-09

**Authors:** Javier Martínez-Gramage, Juan Pardo Albiach, Iván Nacher Moltó, Juan José Amer-Cuenca, Vanessa Huesa Moreno, Eva Segura-Ortí

**Affiliations:** 1Department of Physiotherapy, Universidad Cardenal Herrera-CEU, CEU Universities, 46115 Valencia, Spain; ivan.nacher.molto@gmail.com (I.N.M.); juanjo@uchceu.es (J.J.A.-C.); esegura@uchceu.es (E.S.-O.); 2Embedded Systems and Artificial Intelligence Group, Universidad Cardenal Herrera-CEU, CEU Universities, 46115 Valencia, Spain; juaparal@uchceu.es; 3Triathlon Technification Program, Federación Triatlón Comunidad Valencian, 46940 Manises, Spain; vanessa.huesa@triatlocv.org

**Keywords:** running, kinematics, gait retraining

## Abstract

Background: The running segment of a triathlon produces 70% of the lower limb injuries. Previous research has shown a clear association between kinematic patterns and specific injuries during running. Methods: After completing a seven-month gait retraining program, a questionnaire was used to assess 19 triathletes for the incidence of injuries. They were also biomechanically analyzed at the beginning and end of the program while running at a speed of 90% of their maximum aerobic speed (MAS) using surface sensor dynamic electromyography and kinematic analysis. We used classification tree (random forest) techniques from the field of artificial intelligence to identify linear and non-linear relationships between different biomechanical patterns and injuries to identify which styles best prevent injuries. Results: Fewer injuries occurred after completing the program, with athletes showing less pelvic fall and greater activation in gluteus medius during the first phase of the float phase, with increased trunk extension, knee flexion, and decreased ankle dorsiflexion during the initial contact with the ground. Conclusions: The triathletes who had suffered the most injuries ran with increased pelvic drop and less activation in gluteus medius during the first phase of the float phase. Contralateral pelvic drop seems to be an important variable in the incidence of injuries in young triathletes.

## 1. Introduction

Triathlon is a growing sport with broad participation spanning three disciplines (swimming, cycling, and running) in the same event. This has led to an increase in the incidence of injuries, varying from 37% to 91% in the adult population [1]. In Spain, participation in triathlon has increased by more than 200% among young athletes of school age in recent years (Spanish Triathlon Federation). In the United States, the increase in the participation of children and adolescents in sports, as well as more intense training and specialization at an early age, has contributed to musculoskeletal injuries normally observed in the adult population becoming more common among younger athletes, especially those caused by excessive and repeated use [2]. The risk of musculoskeletal injury in young athletes is related to growth and development which, together with factors such as the rapid increase in the intensity, duration, and volume of physical activity, poor condition, or insufficient sport-specific training, leads to injuries in articular cartilage or other muscle-tendon structures as the result of the exertion of repetitive and excessive stress on the tissues coupled with their lack of adaptation [2,3,4].

Most triathlon injuries are related to the running segment (58–72%) [5,6] and the incidence of such injuries is similar to that of athletics runners [7]. The anatomical area of injury corresponds to the lower extremities [5,7,8], especially the knee [6]. The most frequent types of injuries in running are patellofemoral pain (PFP), iliotibial band syndrome (ITBS), medial tibial stress syndrome (MTSS), Achilles tendinopathy (AT), plantar fasciitis, stress fractures, and muscle strains [9]. The factors related to the development of running-related injuries are multifactorial and diverse; however, it is widely accepted that kinematic alterations during running may also be related [9].

Gait retraining is a clinical intervention based on real-time feedback from wearables which aims to reduce the risk of injury and improve performance and motivation [10]. Current evidence indicates that this technique, alongside traditional therapeutic interventions, should be considered for use in the treatment of injured runners and to address potentially harmful running mechanics in the healthy population [11]. Various authors have shown a decrease in pain in runners with PFP [12,13,14,15] and a 62% reduction in the incidence of injuries in adult athletes [16] as a result of the use of these techniques. Furthermore, Bramah et al. [12] showed an improvement in peak contralateral pelvic drop, hip adduction, and knee flexion after a session of gait retraining, increasing cadence by 10%. Chumanov et al. [17] showed an increase in gluteus medius and gluteus maximus muscle activation associated with an increase in cadence during the final phase of oscillation. To date, there is no evidence on the effect of gait retraining in young triathletes in order to prevent injuries during running.

The objective of this study was to examine the kinematics of the pelvis and activation of the gluteus medius muscle in the float phase to assess their effect on the neuromuscular stability of the pelvis and the incidence of injuries during running in young triathletes over a seven-month observation period after having completed a gait retraining program.

## 2. Materials and Methods

### 2.1. Participants

The participants belonged to the Triathlon Technification Plan in High Performance of the Valencian Community in Spain. The study was approved by the Ethics Committee for Biomedical research at the CEU-Cardenal Herrera University, (reference №: CEI18/137) and is registered as a clinical trial (ClinicalTrials.gov registration №: NCT04221698).

Inclusion/exclusion criteria:

19 triathletes (10 males, 9 females) were enrolled in this study (Table 1). Using G*Power software, we calculated that we would need at least 17 subjects in order to detect a large effect size of 0.8, having a power of 0.87 and a critical alpha of 0.05. This calculation was based on the use of t-tests for two dependent means, to detect differences before and after the gait retraining, as we evaluated the same individuals at two different moments.”

Participants were included if they reported running a minimum of 2 days per week for the 3 months prior with no reported injuries and with their worst pain rated a minimum of 3 out 10 on a numerical rating scale (NRS) for pain (0 = no pain; 10 = worst possible pain) [12]. Participants were excluded if they reported any previous musculoskeletal surgery, neurological impairment, structural deformity in the knee, pain suffered by trauma or sports activities, having stopped running, or having received additional treatment outside of this study.

### 2.2. Data Collection

The data were collected via a self-report questionnaire, similar to previous research in triathletes performed to document the incidence of overuse injuries during the 2018 season and in the post-gait-retraining protocol season (during 2019) [18]. Prior to testing, all the participants performed a 5-min warm-up on a treadmill (HP Cosmos Quasar, Nussdorf-Traunstein, Germany) at their preferred speed [12]. Kinematic and dynamic surface electromyography (sEMG) data were collected over 5 min at 90% of the maximum aerobic power speed (as obtained from the Wasserman protocol) to determine the VO2max [19].

Kinematic data were collected from all participants while running on a treadmill. For the 3D pelvis kinematics, the inertial measurement unit (IMU) was placed in S1 with a belt and raw data was recorded by GSensor and GSTUDIO software version 2.8.16.1. (BTS Bioengineering, Garbagnate, Italy). The validity of the IMU system has previously been shown for the measurement of 3D joint kinematics [20,21]. The 3D pelvis kinematics recorded were the difference in pelvic obliquity for the left and right leg, the tilt, and the rotation. A range of kinematic parameters at both initial contact and midstance were selected for analysis in the sagittal plane from a 2-dimensional video [22]. All the videos were recorded (120 frames per second) with the same camera mounted to a portable tripod and Apple iPad Air tablet computer (Apple Inc, Cupertino, CA). The kinematics angles were measured by using the Hudl Technique video analysis application. Parameters at initial contact included foot-strike pattern, tibial inclination, knee flexion, and forward trunk angles. Peak and midstance angles included dorsiflexion, knee flexion, and forward trunk lean angles. Parameters were selected based on previous research to identify running injury patterns [9,22].

sEMG was simultaneously recorded with the kinematics by placing sEMG electrodes on the gluteus medius [23,24]. The SEMG sensors used in this study were pre-gelled self-adhesive bipolar Ag/AgCl disposable surface electrodes of 20 mm (Infant Electrode, Lessa, Barcelona), with 2 cm interelectrode distance. SEMG electrodes were longitudinally placed on the muscle belly of the dominant leg according to SENIAM recommendations [23]. The EMG signal was recorded simultaneously using a FREEEMG 1000 and EMG Analyzer (BTS Bioengineering, Milan, Italy) that was set to a sampling rate of 1000 Hz per channel, and the signals were band-pass filtered from 20 Hz to 450 Hz. The EMG signals were subsequently full-wave rectified and low pass filtered using a bidirectional, 6th order Butterworth filter with a cutoff frequency of 5 Hz. The root mean square (RMS) in several sub-phases was detected. The IMU sensor detected every event performed; initial contact and toe-off of each foot. Moreover, at the same time, the sEMG signal was recorded, so that the system selected the right and left strides and the different subphases; (first stance phase, first float phase, second stance phase, second float phase).

### 2.3. Retraining Protocol

All participants included completed the 7 months gait retraining program. After baseline assessment, a number of global kinematic contributors to common running injuries were identified and were used for the real-time feedback during the retraining protocol; these were, cadence [12,25], greater peak contralateral pelvic drop (CPD), and trunk forward lean, as well as an extended knee and dorsiflexed ankle at initial contact [9]. Participants were asked to run at a self-selected speed with a 10% increase in their original step rate [11,12,16,25,26]. A modified gait retraining protocol according to Chan et al. [16] was used. In brief, the triathletes participated in four sessions of gait modification over four weeks with one session per week. During the training, the athletes were asked to run at a self-selected speed on a treadmill. Visual biofeedback in the form of a sagittal plane video of the triathlete was displayed on the monitor in front of them (Figure 1).

Participants were instructed by the physiotherapist to modify kinematic variables such as the position of their trunk, contact of the foot with the ground, and knee flexion at initial contact. During the first 5 min, participants were instructed to match their footstep to an audible metronome set to the new step rate which increased their original step rate by 10% [12]. The training time was gradually increased from 15 min to 30 min over the four sessions, and visual/audible feedback was progressively reduced in the last 2 sessions (Table 2). Triathletes were then instructed to maintain their new running pattern during their daily running practice only with their watch cadence as feedback.

### 2.4. Statistical Analysis

With the aim of discovering which variables were more strongly related with the risk of injury among triathletes, we applied machine learning techniques from the artificial intelligence field. Specifically, an ensemble learning method for classification, known as random forests (RF; Breiman, L., 2001) was used to extract the variables that best discriminated between participants who were injured or not in the first period of the study, i.e., before the gait retraining phase. A total of 71 variables were collected from these participants in an excel sheet, although not all these characteristics were selected for the purpose of this present study. Thus, we initially conducted a feature selection protocol to retain only 47 characteristics in order to construct the final dataset as input for the machine learning algorithms. Such variables were selected according to the literature [9,22], to collect data on kinematics, sEMG and running dynamics.

Hence, once the participant data were acquired from the overall observational system, i.e., from the sensors, accelerometers, and video recordings, a raw data set was constructed. After we cleaned this dataset, we converted it into a classification problem for use with machine learning classification techniques. Thus, the problem was added to a supervised learning area’ in which the algorithm tried to learn patterns from data previously labeled for a classification. In our case, a new feature named "injured" was used to classify the triathletes who were injured before the retraining (during 2018) and was our dependent variable.

Once most of the important variables were obtained, we also statistically analyzed them through paired t-tests (with an alpha of 0.05) to compare differences in the pre- and post-test measurements, i.e., before and after the retraining phase. To select the appropriate test, first the normality of the data was checked through the Shapiro–Wilk test (*p* ≥ 0.05). In case normality was not met an equivalent non-parametric alternative to paired t-test is used, in that case the paired samples Wilcoxon test was employed.

## 3. Results

A total of 19 volunteer triathletes initially participated in the study. All of them successfully completed the program and there were no losses to follow-up.

### 3.1. Random Forest Analysis

RF is a well-known algorithm belonging to the family of tree-based methods which yields significant improvements in classification accuracy from large problem sets. It is based on an ensemble of trees which vote for the most popular class [27]. Moreover, trees can capture complex interaction structures with relative bias from among the data and is more competitive than some linear methods [28,29] To develop the model used in this work we used the caret package that integrates the “RandomForest” library [30]. The discriminating ability of the model was assessed by calculating the receiver operating characteristic (ROC) curve to compare different models internally. Additionally, to minimize model overfitting, we used a K-fold cross-validation resampling technique to estimate the efficacy of the model [30], the *K* value was defined at 10. After testing 15 models, the final AUC-ROC was 0.8 (95% CI 0.6–0.9) and the “mtry” parameter (which defines the number of variables randomly sampled as candidates at each split) was 9. The sensitivity was 0.6 (95% CI 0.3–0.8), the specificity was 0.8 (95% CI 0.5–0.9), the NPV (negative predicted value) was 0.7 (95% CI 0.4-0.9) and the MCC (Matthews correlation coefficient) obtained was 0.4. About the values resulted from the confusion matrix, TP (true positives) were 5 and FP (false positives) were 2, nevertheless the TN (true negatives) were 8 and FN (false negatives) 4.

### 3.2. Variable Importance

RF is considered a black-box model because gaining insights on a RF prediction rule is difficult because of the large number of trees generated. Notwithstanding, there is a common approach to extract interpretable information about the contribution of different variables [31] which requires computing so-called variable importance measures to rank the variables (i.e., the features) with respect to their relevance in prediction [32].

Figure 2 shows the variable importance calculation obtained from the RF in this study. The features that appeared were the characteristics that were best able to discriminate the classification of an individual as injured during 2018 or not, i.e., they were the most important global kinematic contributors. Thus, these variables were the objective of this current study. As shown, the pelvic kinematics, knee flexion, ankle dorsiflexion at initial contact, and gluteus medius sEMG were the most important variables in this work.

Once the variables that potentially has the strongest influence on distinguishing injured from non-injured triathletes were identified, we calculated the differences in these variables before and after the retraining program. Table 3 shows which features had the strongest influence on the probability of the triathletes being injured.

Figure 3 shows the difference between the values obtained before and after retraining, as well as their density curves. After retraining, the difference in pelvic obliquity in the right and left limb (A), ankle dorsiflexion in the initial contact (B), contralateral pelvic drop (C, D), and gluteus medius activation during the first phase of flight (E, F) had reduced in almost all of the participants.

Figure 4 shows, in more detail, the differences in pelvic obliquity between participants who were injured during the 2018 season and those who were not injured after retraining. Athletes who were not injured had an average pelvic obliquity of around 2 degrees, while those who were injured had a pelvic obliquity twice that value at 4.22 degrees (A), before retraining (B), after retraining both groups had corrected their pelvic obliquity levels with their mean values homogenizing and coming much closer to zero, thus indicating that they had obtained near symmetry.

Figure 5 shows, the differences in gluteus medius (right) sEMG before and after retraining protocol. Note the increase in activation during the 1st SW.

Figure 6 shows, the differences in pelvic kinematics before and after retraining protocol. Note the reduction in contralateral pelvic drop.

## 4. Discussion

This study identified a number of biomechanical variables that allowed the risk of suffering an injury while running to be detected (Figure 2). To the best of our knowledge, the evidence presented in this work is the first to demonstrate the effect of a gait retraining program in young triathletes in the prevention of injuries. In particular, the triathletes who suffered injuries in the 2018 season had an increased difference in their pelvic obliquity, contralateral fall of the pelvis in the mid-stance phase, increased ankle dorsiflexion during initial contact, and decreased gluteus medius sEMG readings in the first phase of float (Figure 3). We found that differences in pelvic obliquity and contralateral pelvic drop were the most important predictive variables of injury when classifying triathletes as injured or non-injured. These kinematic patterns coincide with the results obtained in previous studies [9,12], except that our study was carried out in a young population for which no similar data is yet available.

Various authors [12,33] have hypothesized that the delay in gluteus medius activation during the stance phase of running could alter neuromuscular control in the hip and pelvis, thus facilitating the loss of stability in the frontal plane. In this study, a significant increase in gluteus medius activation was achieved during the first phase of float. This increase occurred prior to the strike of the contralateral foot, facilitating neuromuscular control in the frontal plane, improving both the difference in the range of obliqueness in each limb and in the fall of the contralateral pelvis. In agreement with other studies that also obtained positive results [12,25,26], this increase in muscle activation seemed to be the result of the increased cadence established during the gait retraining program (by 10% compared to their cadence from the initial assessment), but did not seem to be related to an increase in pelvis stability, making this work the first to show this effect. Bonacci et al. demonstrated that movement patterns in triathletes during the transition from cycling to running are altered at the neuromuscular level [34]. Even in veterans and highly trained triathletes, there is altered muscle recruitment after cycling, which can lead to tibial stress fractures from overuse which may be associated with increased bone load caused by impaired neuromuscular control [35].

Various authors have pointed out that the knee is the most common location for acute and overuse problems in triathletes, followed by the lower leg, lumbar area, and shoulder [5]. Overuse was the reported cause in 41% of the injuries, two-thirds of which occurred during running [36]. Many triathletes continue to train, albeit on a modified routine, after an injury and so further injurious exposures may occur [7]. Defective movement patterns have previously been associated with injuries and pain, although there is no homogeneity between the running pattern and its location. Studies have shown that strengthening exercises alone do not alter these patterns and so a different approach to treatment targeting the motor level is necessary to effect these changes [26]. Therefore, movement retraining, while still adhering to basic principles of motor control, should be part of the intervention skill set [15]. Our study echoes these results but, unlike the studies published to date, focused on young triathletes. Thus, the concepts discussed above could help explain the decrease in the number of injuries produced after the gait retraining program.

Although, one of the most commonly used statistical learning models for discriminant analysis is logistic regression, we were concerned that this technique would only capture linearities in data. However, because RF is a non-parametric machine learning technique, it has additional, powerful capabilities for this type of analysis. This is why this technique is preferred in many medical applications, both for its excellent prediction performance but also its ability to identify important variables [37]. Thus, we decided to implement tree techniques such as RF in this current work. These techniques are simple and powerful machine learning models, that can generate a set of highly interpretable conditions that are straightforward to implement [31].

### Limitations of the Study and Future Activities

One of the limitations of the study is the lack of a control group. However, all the included triathletes fulfilled homogeneous inclusion criteria running a minimum of two days per week for the three months prior with no reported injuries and with their worst pain rated a minimum of 3 out 10 on a numerical rating scale (NRS) for pain (0 = no pain; 10 = worst possible pain). A second limitation of this work is the sample size, since this only allows us to obtain clues about what we are investigating although encourages us to continue working in this line as the results seem very promising. However, we must also take into account that obtaining data on high-performance athletes is quite difficult since it is a very small population and therefore the sample can never be too large. Nevertheless, it is also true that the sample, despite being small, is quite homogeneous. This allows us to think that the conclusions of the research could be generalized to other athletes, who will have very similar characteristics to the sample we are working on. On the contrary, is difficult to profit all the potential of the present artificial intelligence techniques that bring a new framework to study the data, such models need from large datasets. The future directions of this work should be addressed towards the application of these results in the field of training of young triathletes to reduce running injuries. Future research should determine biomechanical running patterns that indicate a lower incidence of injury in young athletes.

## 5. Conclusions

This study identified a number of scaled and related variables based on their importance in preventing injuries during running. In particular, we found an increase in the obliquity of the pelvis, fall of the contralateral pelvis, the extension of the knee, dorsiflexion of the ankle in the initial contact, and less activation of the gluteus medium during the first phase of float in triathletes who suffered injuries. After the gait retraining program, the number of injuries was reduced by improving the neuromuscular stability of the pelvis of these athletes, thus providing an easy way to assess and readjust their running style in clinical practice.

## Figures and Tables

**Figure 1 sensors-20-06388-f001:**
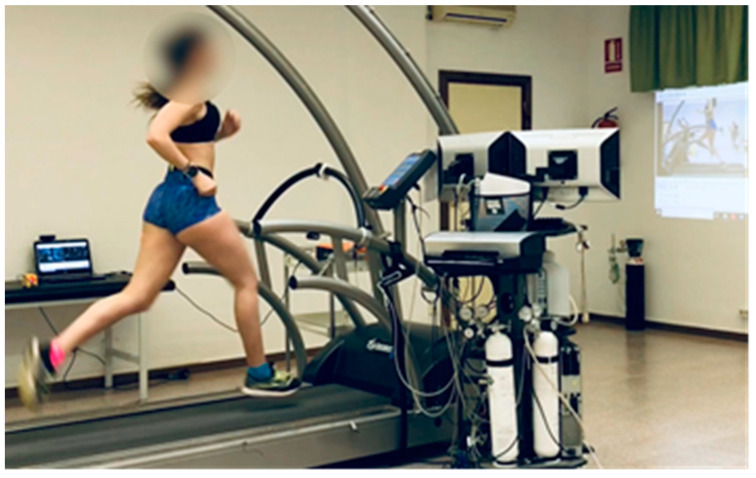
Visual real-time biofeedback during the retraining protocol.

**Figure 2 sensors-20-06388-f002:**
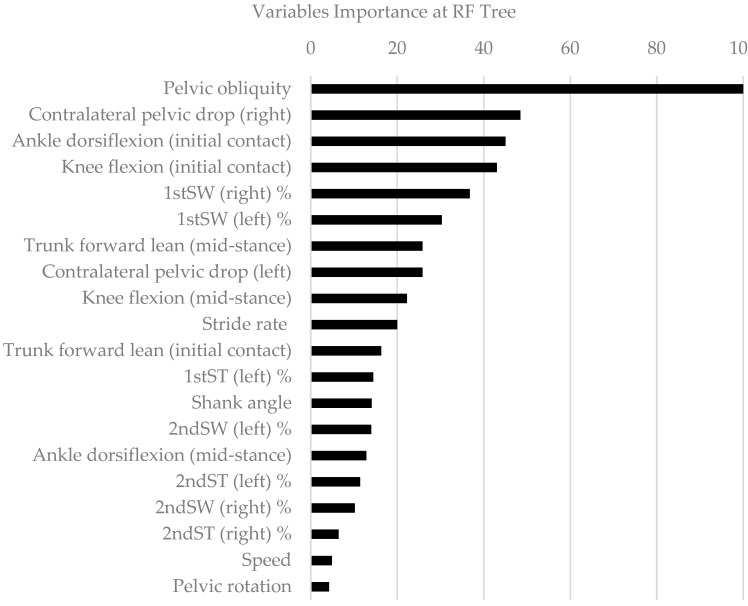
Importance of the variables, scaled according to the “varImp” method in the caret R library for the complete data set.

**Figure 3 sensors-20-06388-f003:**
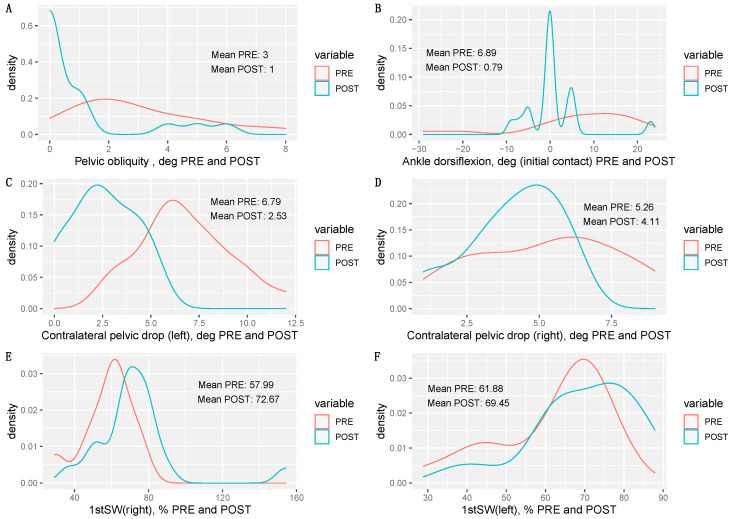
Density plots showing the differences between pre- and post-values obtained before and after the retraining phase. Higher density values on the ordinate axis point out which are the most probable values on the abscissa axis. The difference in pelvic obliquity in the right and left limb (**A**), ankle dorsiflexion in the initial contact (**B**), contralateral pelvic drop (**C**,**D**), and gluteus medius activation during the first phase of flight (**E**,**F**).

**Figure 4 sensors-20-06388-f004:**
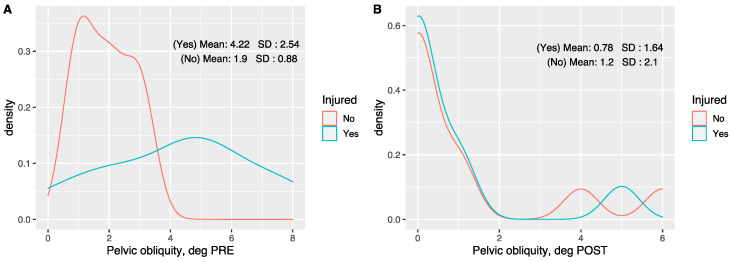
Density plots comparing the differences between injured and non-injured triathletes in terms of the degree of pelvic obliquity between the (**A**) pre-retraining; (**B**) and post-retraining phase values. Higher density values on the ordinate axis point out which are the most probable values on the abscissa axis.

**Figure 5 sensors-20-06388-f005:**
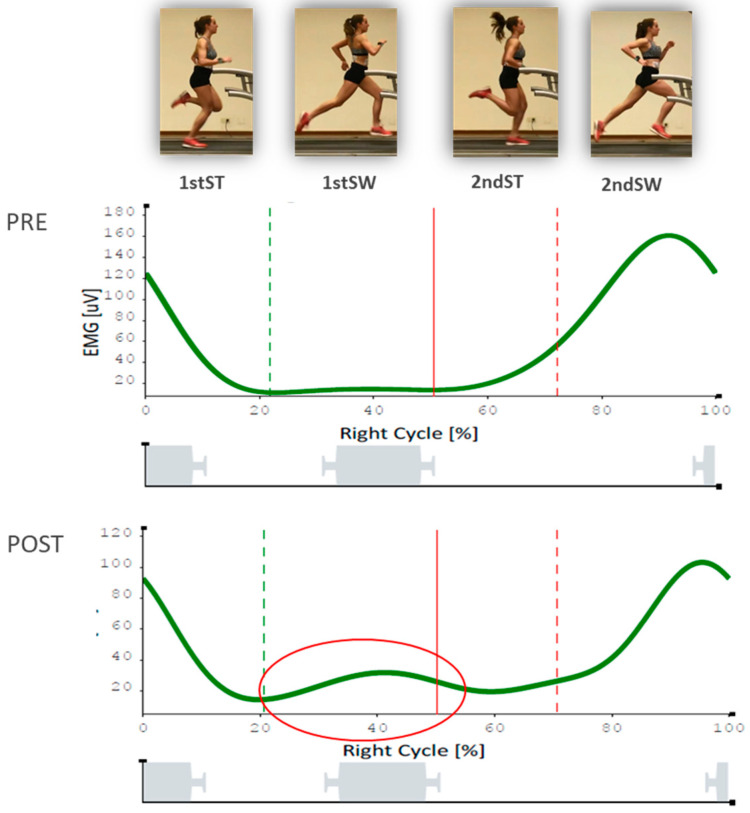
Gluteus medius (right) sEMG plot pre and post.

**Figure 6 sensors-20-06388-f006:**
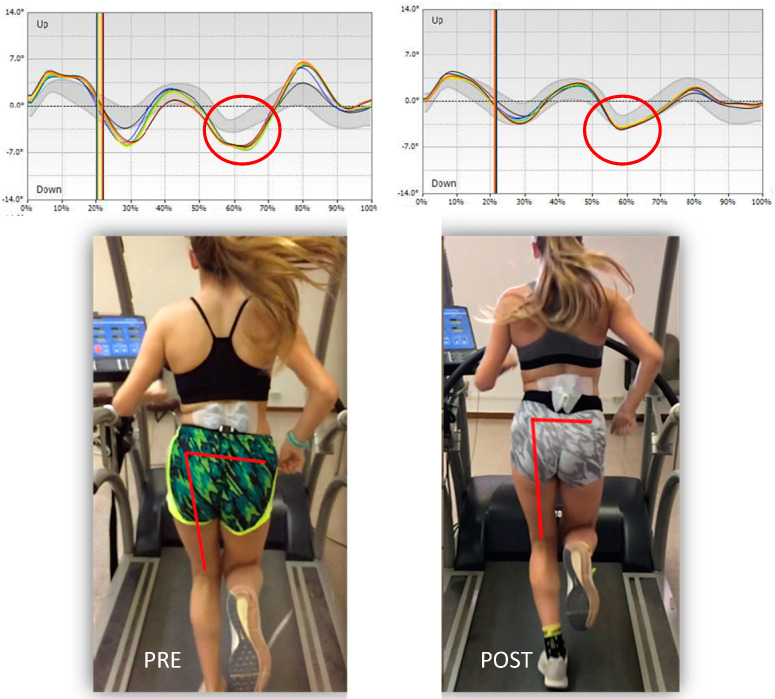
Pelvis kinematics before and after retraining protocol.

**Table 1 sensors-20-06388-t001:** Participant characteristics ^a^.

	Healthy (*n* = 10)	Injured (*n* = 9)
Age	14.8 ± 1.9	14.4 ± 1.7
Weight, kg	52.7 ± 7.9	56.1 ± 10.9
Height, cm	167.1 ± 8.1	169.1 ± 9.6
Body mass index, kg/m^2^	18.8 ± 1.6	19.4 ± 1.8
Years in competition	7.2 ± 1.7	6.8 ± 1.8
Training hours per week	19.2 ± 5.7	17.9 ± 5.1

^a^ Values are presented as the mean ± SD.

**Table 2 sensors-20-06388-t002:** Gait retraining protocol ^a^.

	VSP	A.M	WCd	Time Session
Session 1 (min)	10′	5′	−	15′
Session 2 (min)	15′	5′	−	20′
Session 3 (min)	10′	−	15′	25′
Session 4 (min)	−	−	30′	30′

^a^ Training time and biofeedback time arrangement. VSP, video sagittal plane; AM, audible metronome; WCd, watch cadence.

**Table 3 sensors-20-06388-t003:** Analysed variables ^a^.

	PRE	POST	*p* Value
Stride rate, steps/min	174.4 ± 8.3	181.4 ± 7.7	0.00 ^b^
Speed, km/h	15.9 ± 1.7	16.5 ± 2.3	0.2
Run cycle, sec	0.69 ± 0.0	0.66 ± 0.0	0.00 ^b^
Pelvic obliquity, deg	3 ± 2.1	1 ± 1.8	0.01 ^b^
Pelvic tilt, deg	9.4 ± 1.2	9.9 ± 2.3	0.41
Pelvic rotation, deg	21 ± 5.7	19.7 ± 5.2	0,13
Trunk forward lean, deg (initial contact)	7.2 ± 5.3	3.6 ± 2.2	0.00 ^b^
Knee flexion, deg (initial contact)	18.8 ± 6.5	22.1 ± 3.1	0.03 ^b^
Ankle dorsiflexion, deg (initial contact)	6.8 ± 13.3	0.7 ± 6.6	0.03 ^b^
Shank angle, deg (initial contact)	10.6 ± 3.9	5.4 ± 3	0.00 ^b^
Knee flexion, deg (mid-stance)	44.4 ± 5.3	36.8 ± 9.6	0.00 ^b^
Ankle dorsiflexion, deg (mid-stance)	19.1 ± 7.9	16.6 ± 4	0.28
Trunk forward lean, deg (mid-stance)	11.1 ± 3.8	9.7 ± 2.9	0.24
Contralateral pelvic drop (left), deg	6.7 ± 2.3	2.5 ± 1.6	0.00 ^b^
Contralateral pelvic drop (right), deg	5.2 ± 2.4	4.1 ± 1.5	0.04 ^b^
1stST (right), %	52.2 ± 15.7	56.1 ± 26.6	0.6
1stSW (right), %	57.9 ± 13.4	72.6 ± 23.6	0.01 ^b^
2ndST (right), %	69.7 ± 14.3	96.6 ± 102.7	0.65
2ndSW (right), %	63.4 ± 26.8	61.5 ± 18	0.68
1stST (left), %	66.2 ± 57.4	46.9 ± 24.6	0.06
1stSW (left), %	61.8 ± 14.2	69.4 ± 13.4	0.01 ^b^
2ndST (left), %	80.7 ± 42	67 ± 23.8	0.46
2ndSW (left), %	58.7 ± 11.9	78.6 ± 40.8	0.08

^a^ Values are presented as the mean ± SD using paired t-tests. 1stST, first stance phase; 1stSW, first float phase; 2ndST, second stance phase; 2ndSW, second float phase. ^b^ Statistical significance was set at *p* ≤ 0.05.

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
