# Peer review of "A Random Forest Machine Learning Framework to Reduce Running Injuries in Young Triathletes"

_sensors, 2020, doi:10.3390/s20216388_

Round 1
Reviewer 1 Report
Why was the effect size of 0.75 used in this study? What was the test used in G-power to calculate the sample size?
Line 108, how was the validity of the questionnaire assessed?
How was the sEMG and IMU recordings synchronized?
sEMG recording protocol lines 128-134:
The proper standards for reporting sEMG data must be used:
https://www.sciencedirect.com/science/article/pii/S105064111400042X
For example, the shape and type of the electrodes, their IED, electrode montage (Monopolar, Single-differential, or Double differential), where was the GND or reference electrodes connected, how were the electrodes located on the muscles and a critical issue, how was the muscle IZ/tendon avoided?
Please provide plots of the sEMG and IMU recording in the paper.
How was the quality of the sEMG recordings monitored, and how were the outliers avoided?
How was the power-line interference removed from the sEMG data?
Statistical analysis
Lines 164-167
Please report the descriptive statistics in the injured and non-injured groups as well in Table 1.
How many subjects were injured in the first period of the study? Please specify.
Line 171,
What are the 47 characteristics? Please specify.
Lines 180-184, Did the authors use the normality test for pre and post variables?
Line 197, what is the value of K in K-fold cross-validation?
I do not see any performance measure of the classification. It is required to report the parameters Sensitivity, Specificity, PPV, and MCC and AUC. The mean and DS of these values over the test folds must be reported. The cross-validated confusion matrix must be added to the paper, and the performance indices must be reported with their CI 95% (also for AUC) following the STARD and TRIPOD recommendations.
The quality of Figs 3and 4 must be improved.
Line 290,
Did the authors try logistic regression? It could provide the probability of belonging to the injured class as one of its advantages.
What is the limitation of the current study and future activities?
Author Response
Please, see the attachment

Reviewer 2 Report
This report provided an important work on preventing injuries during running for young triathletes, and the results showed great improvements. I think the paper can be accepted, but some problems should be addressed before the publication.
1.The quantity of figures should be improved; low resolution makes them not very readable.
2.Many key variables were identified and promoted. A simple description about their roles in running would be better to understand their importance.
3.The ordinate in Figures 3,4 used density. Its meaning is missing.
4. Fig.4 Change injury into injured.
Author Response
Please, see the atachment

Round 2
Reviewer 1 Report
The paper is suitable for publication.
Reviewer 2 Report
The revison work is great, and the paper can be accpeted for publication.
A little problem about the paper type. In the system, it is grouped into Case reprot, but is shown as Acticle in the manuscript.